# β-sheet stabilization of the island domain underlies ligand-induced LRR-RP activation of plant immune signaling

Simon Snoeck [1] ✉, Lisha Zhang [2], Valentin Studer [1], Gijeong Kim [1], Álvaro D. Fernández-Fernández [1], Thorsten Nürnberger[2] & Cyril Zipfel [1,3] ✉

Leucine-rich repeat (LRR) receptor kinases (RKs) and receptor proteins (RPs) are important classes of plant pattern recognition receptors (PRRs) activating pattern-triggered immunity. While both classical and AI-based structural approaches have recently provided crucial insights into ligand-LRR-RK binding mechanisms, our understanding of ligand perception by LRR-RPs remains limited. Here, we employed an AI-based approach to reveal a ligand-binding mechanism shared by the Arabidopsis LRR-RPs RLP23 and RLP42 – the PRRs for the short peptide ligands nlp20 and pg13, derived from NECROSIS- AND ETHYLENE-INDUCING PEPTIDE 1-like proteins (NLPs) and fungal endopolygalacturonases (PGs), respectively. Additionally, we investigated the larger and more complex binding interface of RLP32 – the PRR for proteobacterial TRANSLATION INITIATION FACTOR 1 (IF1), a folded protein ligand that requires its tertiary structure for recognition. Finally, we describe a mechanistic role of the ID for co-receptor recruitment conserved across LRR-RPs. Together, our results shed light on the ligand-binding mechanisms and receptor complex formation of LRR-RPs, opening avenues for their engineering for crop disease resistance.

Effective sensing and response to biotic stress is crucial for plant health. Plant immune recognition is mediated by intracellular nucleotide-binding and leucine-rich repeat receptors (NLRs) and cell-surface pattern recognition receptors (PRRs). PRRs are either receptor kinases (RKs) or receptor proteins (RPs). The extracellular ligand-binding domains of PRRs have diverse sizes and architectures[1], with the most common ectodomain being composed of leucine-rich repeats (LRRs)[2]. RPs do not contain a cytoplasmic kinase domain and rely for signal transduction on a constitutive association with the LRR-RK SUPPRESSOR OF BRASSINOSTEROID INSENSITIVE 1-ASSOCIATED KINASE 1-INTERACTING RECEPTOR-LIKE KINASE 1/EVERSHED (SOBIR1/EVR) and ligand-induced association with the co-receptor BRASSINOSTEROID INSENSITIVE 1-ASSOCIATED KINASE 1/SOMATIC EMBRYOGENESIS RECEPTOR KINASE 3 (BAK1/SERK3)[3,4]. Although

LRR-RPs were among the first disease resistance genes cloned[5], their structure-function mechanisms are poorly understood relative to NLRs and RK-type PRRs. The cryo-EM structure of the LRR-RP RESPONSE TO XEG1 (RXEG1) is to date the sole resolved structure of an LRR-RP-type PRR[6]. Besides RXEG1, certain subdomains of LRR-RPs and residues were pinpointed as crucial for receptor functionality through genetic experiments and comparative genomics but only provided limited mechanistic insights[4,7–17]. A better understanding of LRR-RP structure-function mechanisms can inform PRR engineering and hence create opportunities to enhance plant disease resistance against a broad spectrum of pathogens, enabling durable agricultural practices[18–20].

Most LRR-RPs embed one or more loopout domains in their extracellular domain (Supplementary Fig. 1)[1,4,8]. N-terminal (NT)

[1]Institute of Plant and Microbial Biology, Zurich-Basel Plant Science Center, University of Zürich, Zürich, Switzerland. [2]Center of Plant Molecular Biology (ZMBP), University of Tübingen, Tübingen, Germany. [3]The Sainsbury Laboratory, University of East Anglia, Norwich Research Park, Norwich, United Kingdom. ✉e-mail: simon.snoeck@uzh.ch; cyril.zipfel@uzh.ch

loopout domains can be embedded within both the NT capping domain of the LRRs, and/or the NT LRRs itself, whereas the island domain (i.e., C2-domain) is consistently positioned N-terminally to the last set of four LRRs of the LRR domain (C3 domain)[4,8]. The fixed position of the ID suggests a conserved function[8]. Additionally, the presence of an ID is shared with LRR-RKs of the subgroup Xb, which were recently suggested to share a common origin and mechanistic role with LRR-RPs[8]. Using chimeric receptors that leverage closely related paralogues, the requirement of the respective ID for LRR-RP function was earlier demonstrated for tomato Ve1, Arabidopsis RLP42 and cowpea INCEPTIN RECEPTOR (INR)[7,9,11]. Moreover, an RLP42 mutant with an amino acid (AA) substitution in the ID (E696K) had impaired ligand binding and function[9]. RXEG1 contains two loop-out regions, an NT loopout and an ID. While the inner surface of the LRR domain wraps the ligand GLYCOSIDE HYDROLASE 12 (GH12) protein (named XEG1), the interaction between XEG1 and RXEG1 is primarily mediated by both loopouts[6]. However, the IDs of LRR-RPs with defined ligands vary strongly in size and composition and NT loopouts are not uniformly present[4]. For example, Arabidopsis RLP23 and RLP42, and tomato Cf-2 and Cf-5 do not contain an NT loopout. Besides, characterized LRR-RP ligands range from short peptides like pg13 and nlp20 to larger tertiary folded ligands such as XEG1 and proteobacterial TRANSLATION INITIATION FACTOR 1 (IF1). Overall, this suggests diverse uncharacterized ligand-binding mechanisms across LRR-RPs[4].

In addition to multiple conserved residues across LRR-RPs and LRR-RK-Xb members within the C3 domain[8], the RXEG1 complex structure indicates that the role of its ID extends to association with the co-receptor BAK1[6]. Intriguingly, conformational changes of the C-terminal (CT) ID from α-helix to an antiparallel β-sheet were described upon complex formation[6]. LRR-RP IDs can be categorized based on two conserved motifs in the CT part of the ID that are strongly conserved across LRR-RPs, K-$x_5$-Y and Y-$x_8$-KG[4,8]. Moreover, both motifs share a lysine (K) residue[4]. The corresponding RXEG1 residue K807 is positioned within the linking loop of the two parallel strands forming the CT ID β-sheet of RXEG1 and was shown to be crucial for RXEG1-BAK1 interaction[6]. Taken together, we hypothesize a

conserved functional role for the CT part of the ID of LRR-RPs in ligand-induced BAK1 recruitment through β-sheet formation.

We recently leveraged AI-based structural modeling to identify the conserved binding pockets of the LRR-RK MALE DISCOVERER 1-INTERACTING RECEPTOR-LIKE KINASE 2 (MIK2) with the S-x-S motif present in the large and diverse family of SERINE RICH ENDOGENOUS PEPTIDES (SCOOP) phytocytokines[21]. Notably, two subsequent studies using X-ray and cryo-electron microscopy (cryo-EM) structural approaches confirmed the same residues for interaction of the S-x-S motif of SCOOP12 with MIK2[22,23]. Considering the technical challenges, time, effort, and cost of classical structural determination approaches[24], establishing AI-based structural modeling for LRR-RPs could further help to elucidate LRR-RP structure-function mechanisms.

Here, we extended the use of AI-based protein predictions from LRR-RKs to LRR-RPs, a rapid and inexpensive alternative to classical structure-based approaches. In doing so, we revealed a LRR-RP ligand-binding mechanism shared by Arabidopsis RLP23 and RLP42 through the adoption of an antiparallel β-sheet conformation by the NT part of their ID and a β-strand interaction between their corresponding ligands. In addition, we characterized the diverse and complex AI-predicted binding interface of RLP32. Finally, we functionally validated a conserved mechanistic role across LRR-RPs of the CT ID for co-receptor recruitment.

## Results

### The NT parts of the RLP23 and RLP42 IDs are predicted to adopt an antiparallel β-sheet conformation and interact through a β-strand with their ligands

We used AlphaFold 3 (AF3) to predict the structural interaction between multiple characterized ligands and their corresponding LRR-RP ectodomains, with and without the coreceptor BAK1 (Fig. 1a)[21,25]. The conserved nlp20 motif sensed by Arabidopsis RLP23 was earlier identified and thoroughly characterized through leveraging a multitude of diverse fragments of NECROSIS- AND ETHYLENE-INDUCING PEPTIDE 1 (Nep1)-LIKE PROTEINS (NLPs) and an alanine-scanning mutagenesis approach[26]. Likewise, the structural motif of fungal

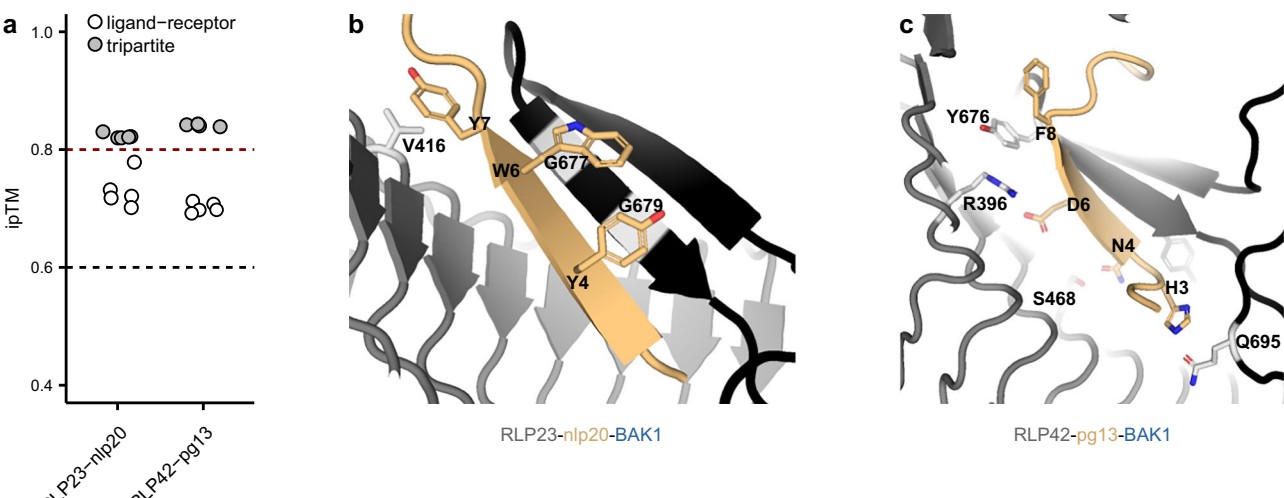

**Fig. 1 | The N-terminal parts of the IDs of RLP23 and RLP42 are predicted to adopt a β-strand conformation and interact through a β-strand with their respective ligands. a** AF3 predicts a high interface predicted Template Modeling (ipTM) for RLP23 and RLP42 in complex with their respective ligand and the co-receptor BAK1 (tripartite). AF3 guidelines state that ipTM values higher than 0.8 represent confident high-quality predictions. ipTM values between 0.6 and 0.8 are within a gray zone where predictions could be correct or incorrect. The AF3 cutoffs are depicted with a red and black dotted line. **b**, **c** Structural representations of the

tripartite complexes of RLP23-nlp20-BAK1 and RLP42-pg13-BAK1. The ID is highlighted in black, other LRR-RP domains are depicted in dark gray. The ligand is depicted in yellow, labeled residues were earlier shown to be essential for functionality from the ligand side. Highlighted residues in light gray were mutagenized from the receptor side in this study[9,26]. Pdb files can be found in Supplementary Data 1 and the predicted local distance difference test (pLDDT) scores can be found in Supplementary Fig. 2.

endopolygalacturonases (PGs) sensed by Arabidopsis RLP42 was identified and characterized[9]. Hence, similar to the conserved SCOOP S-x-S motif essential for SCOOP function, residues of functional importance on the ligand side of the interaction help to evaluate and interpret AI-based predictions of putative ligand-binding interfaces of LRR-RPs with high interface predicted Template Modeling (ipTM) scores (Fig. 1)[9,21,26].

The NT part of the IDs of RLP23 and RLP42 are predicted to adopt an antiparallel β-sheet conformation and interact through a β-strand with nlp20 and pg13, respectively (Fig. 1, and Supplementary Fig. 2). Considering RLP23-nlp20, Böhm et al. 2014 revealed that the nlp20 residues Y4 and W6 are essential for elicitor activity[26]. Notably, the bulky side chains of Y4 and W6 are predicted to form hydrophobic interactions with the peptide bonds of G679 and G677, thereby stabilizing the antiparallel β-sheet conformation (Fig. 1b, and Supplementary Fig. 3a). Additionally, the nlp20 residue Y7 is predicted to form a hydrophobic interaction with V416 embedded within the LRRs of RLP23. RLP42-pg13-BAK1 prediction showed that Q695, S468, R396, and Y876 of RLP42 interact with the pg13 residues H3, N4, D6, and F8 respectively (Fig. 1c, Supplementary Fig. 3b), which are essential for pg13 eliciting activity[9].

### Hindering the interaction between the ligand and both the NT part of the ID and the C1 LRR domain affects RLP23 and RLP42 functions

To test experimentally the predicted interaction interfaces of RLP23 and RLP42 with their respective ligands, we created constructs with single and double AA substitutions ranging from conservative to drastic side chain changes. We expressed wild-type Arabidopsis RLP23 or RLP42 and their respective variants in *Nicotiana benthamiana*, which lacks both receptor orthologues and is insensitive to nlp20 and pg13[9,10,18]. Western blot analysis demonstrated that RLP23 and RLP42 variants accumulate at comparable protein levels with the respective wild-type receptors except for RLP42[R396E], which appears to be expressed at a lower level (Supplementary Fig. 4b). Incorrect localization and misfolding are unlikely for RLP23 variants as relative to mock treatments, none of them are completely abolished for all three immune signaling outputs tested (Fig. 2a, c, e). Considering RLP42 variants (including RLP42[R396E]), confocal microscopy confirmed similar localization relative to wild-type RLP42 (Supplementary Fig. 5a). Subsequently, the capacity of wild-type and variant receptors to perceive their respective ligands and induce immune signaling was measured by ligand-induced reactive oxygen species (ROS) production, increase in cytoplasmic $Ca^{2+}$ concentrations and ethylene production (Fig. 2, and Supplementary Fig. 6).

Considering RLP23, because the R groups of G677 and G679 are facing Y4 and W6, respectively, we disrupted the stabilization by the two bulky hydrophobic acids of nlp20 through the introduction of steric hinderance (Fig. 1b). The single conservative AA changes G679A and G677V in the NT part of the ID consistently reduced RLP23 function independent of the immune signaling outputs measured (Fig. 2a, c, e). Moreover, the corresponding G677V/G679A double mutation reduced ethylene production and completely abolished ROS and $Ca^{2+}$ bursts. The more drastic single AA changes of G677 also resulted in reduced (G677E, G677F) or abolished (G677R) ROS bursts (Supplementary Fig. 6). Hence, the AA changes providing steric hindrance or disrupting hydrophobic interaction between nlp20 and the NT part of the RLP23 ID seem sufficient to hinder the predicted interaction. The single V416A change, embedded within the C1 LRRs of RLP23 and predicted to interact with Y7 of nlp20 through a hydrophobic interaction, significantly reduced $Ca^{2+}$ burst and ethylene production (Supplementary Fig. 1). A more drastic AA change, V416R, was required to further reduce ROS production (Fig. 2 and Supplementary Fig. 6),

whereas the double mutation V416A/G677V abolished ROS and ethylene productions. Thus, besides G677 and G679, V416 contributes to RLP23 function.

Considering the RLP42-pg13-BAK1 prediction, the single AA changes R396A and Y676A/L, respectively, disturbed an ion interaction with D6 and the hydrophobic interaction stacking with F8 of the ligand pg13 (Fig. 1c). This was reflected by abolished ROS and $Ca^{2+}$ bursts as well as ethylene accumulation for both RLP42 variants (Fig. 2b, d, f). Considering R396, removal of charge through the introduction of an alanine was sufficient to block RLP42 function. RLP42[S468L] significantly reduced pg13-induced $Ca^{2+}$ burst and abolished ROS and ethylene productions, strengthening the proposed importance of its polar interaction with pg13 N4. Finally, all three immune signaling outputs were reduced but not abolished for the variant RLP42[Q695A], where Q695 that putatively interacts with pg13 H3 was mutated. Notably, the key residues identified here for ligand-binding by RLP42 are conserved in RLP40 and therefore could theoretically not be pinpointed through a chimeric approach[9]. Nonetheless, earlier identified residues impacting RLP42 function through such chimeric approach could be reevaluated in the light of the AF3 prediction of the RLP42 receptor complex (Supplementary Fig. 7)[9,27]. Overall, these earlier characterized single AA changes of RLP42 correlate with and thus further validate the AF3-predicted structure of the RLP42-pg13-BAK1 complex[9].

In general, physiological data of variants on both the ligand side and the receptor side strengthens the computational prediction of an interaction of RLP23 and RLP42 with their respective ligands through an antiparallel β-sheet.

### Hindering the putative interaction between the conserved five stranded β-barrel of IF1 and RLP32 affects RLP32 function

A multitude of diverse IF1 fragments were previously tested for eliciting activity but only a minimal N-terminal deletion of six AA residues showed significant activity (I7-R72)[28]. Hence, unlike NLPs and PGs, IF1 elicitor activity could not be assigned to a small epitope suggesting the importance of its tertiary structure for its sensing by RLP32, and consequently a putative diverse ligand-binding mechanism.

AF3 was thus leveraged to predict the structural interaction between the RLP32 ectodomain and IF1, with and without the co-receptor BAK1 (Fig. 3a, and Supplementary Fig. 2)[21]. We used the full AA sequence of IF1 for RLP32 receptor complex predictions. The AF3 prediction showed IF1 in a highly similar conformation as bound to the 30S ribosomal subunit, *i.e.* in an oligonucleotide/oligosaccharide-binding (OB) fold, characterized by a five stranded β-barrel with short alpha helix (Fig. 3a)[29]. The six N-terminal AAs of IF1 were predicted with very low confidence suggesting flexibility (Supplementary Fig. 2f), aligning with the observation of non-requirement for IF1 sensing[28]. The structured IF1 OB fold forms an extensive interaction interface with the LRR domain and the NT part of the ID domain of RLP32, mediated by electrostatic and hydrophobic interactions. K152 and E244 of RLP32 form ionic interactions with E8, E56, and R66 of IF1. In contrast to RLP23 and RLP42, the NT part of the ID domain of RLP32 does not form antiparallel β-sheet conformation; rather, F653 is buried inside of the hydrophobic pocket of IF1. Intriguingly, E244, L340 and F653 are completely and K152 partly conserved within putative RLP32 orthologues earlier identified through synteny[18].

To test the predicted interaction interfaces of RLP32 experimentally, we created mutant variants predicted to disrupt the interaction between RLP32 and IF1. We expressed wild-type Arabidopsis RLP32 and the respective variants in *N. benthamiana*, which lacks an RLP32 orthologue and is insensitive to IF1[18,28]. Western blot analysis and confocal microscopy demonstrated that most RLP32 variants accumulated comparatively to wild-type RLP32 (Supplementary Fig. 4c, and

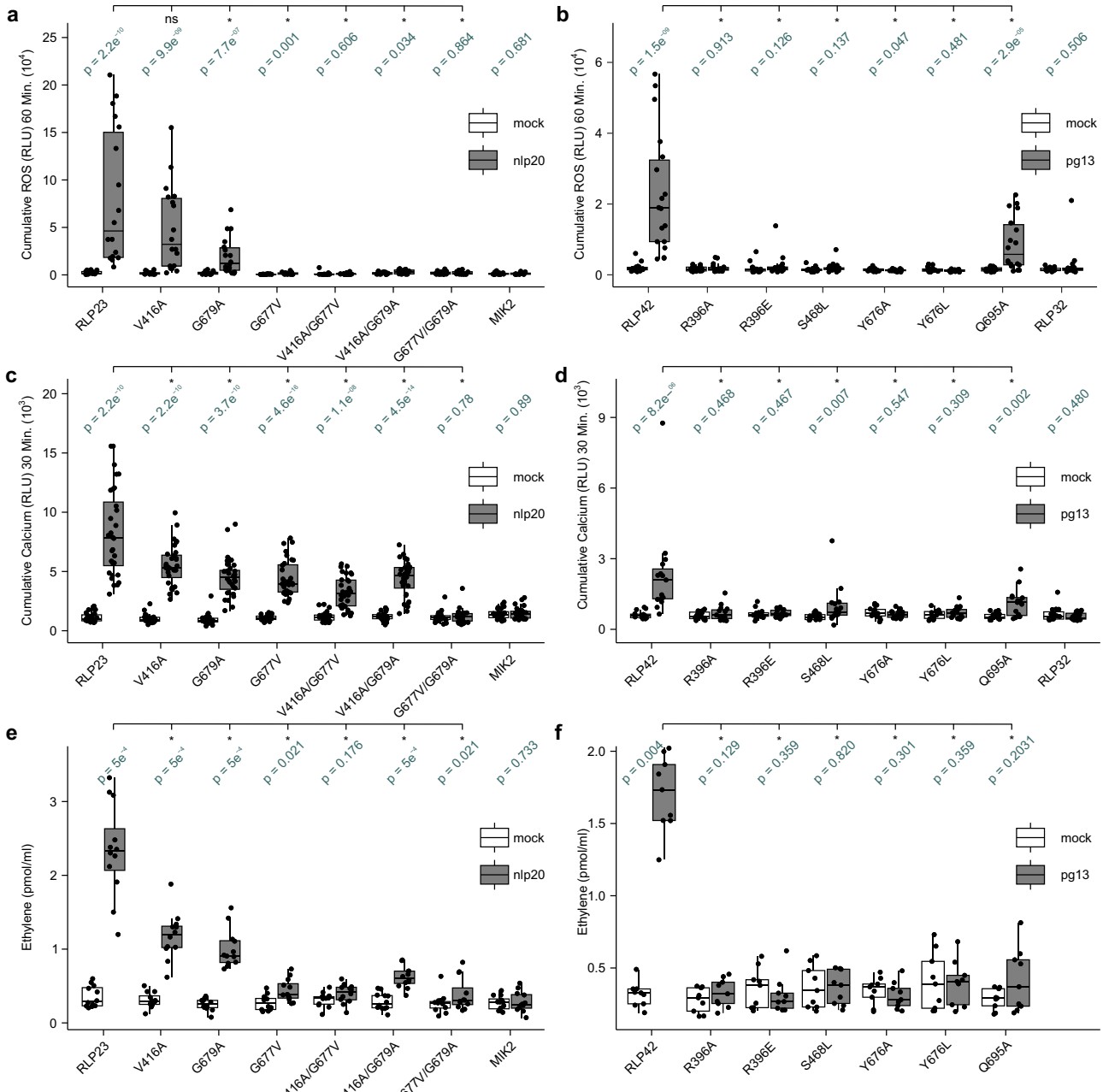

**Fig. 2 | Single and double AA changes within the predicted ligand-binding interfaces affect the functionality of RLP23 (left) and RLP42 (right). a–f** Diverse responses in *N. benthamiana* post heterologous expression of receptor and receptor variants and subsequent treatment with mock treatment (white) or the respective ligand (1 µM nlp20 or 1 µM pg13, gray). Each biological replicate is represented by at least three technical replicates. Box plots indicate the median and the interquartile range (IQR), the whiskers extend to the most extreme data points within 1.5 times the IQR. Significance was tested by performing non-parametric two-sided Wilcoxon-Mann-Whitney tests between both mock and ligand for each receptor (variant) (depicted in green), as well as the ligand-treated WT receptor vs specific variants, without adjustments for multiple comparisons. The asterisks indicate a significant difference of $p < 0.05$. **a, b** ROS production (4–60 min) in cumulative relative luminescence units (RLUs). Six independent biological replicates ($n = 6$ plants) were performed. **c, d** Cytosolic $Ca^{2+}$ concentrations (3–30 min) in cumulative RLUs. At least five independent biological replicates ($n = 8$ (RLP23) or 5 (RLP42) plants) were performed. **e, f** Ethylene accumulation after 4 h treatment, dots represent individual data points from at least three independent experiments ($n = 4$ (RLP23) or 3 (RLP42) plants).

Supplementary Fig. 5b). RLP32 variant misfolding and incorrect localization is unlikely as confocal microscopy confirmed similar localization relative to wild-type RLP32 (Supplementary Fig. 5b).

The F653E mutation results in complete abolishment of ROS and ethylene productions and a significant reduced cytosolic $Ca^{2+}$ burst (Fig. 3). L340 is part of the RLP32 LRR domain and is proximal to the IF1 β-barrel. The L340R mutation presumably leads to steric hinderance for IF binding and results in significantly reduced immune responses

(Fig. 3b–d). By introducing electrostatic repulsion, the E244R mutation resulted in significantly reduced ROS and ethylene productions, and K152E resulted in a significant reduction of $Ca^{2+}$ burst although ROS and ethylene productions were not significantly affected.

Overall, the physiological data obtained from both the created RLP32 variants and earlier published IF1 variants supports the AF3-predicted extensive interaction interface between the five stranded β-barrel and the LRR domain and NT part of the ID domain of RLP32[28,29].

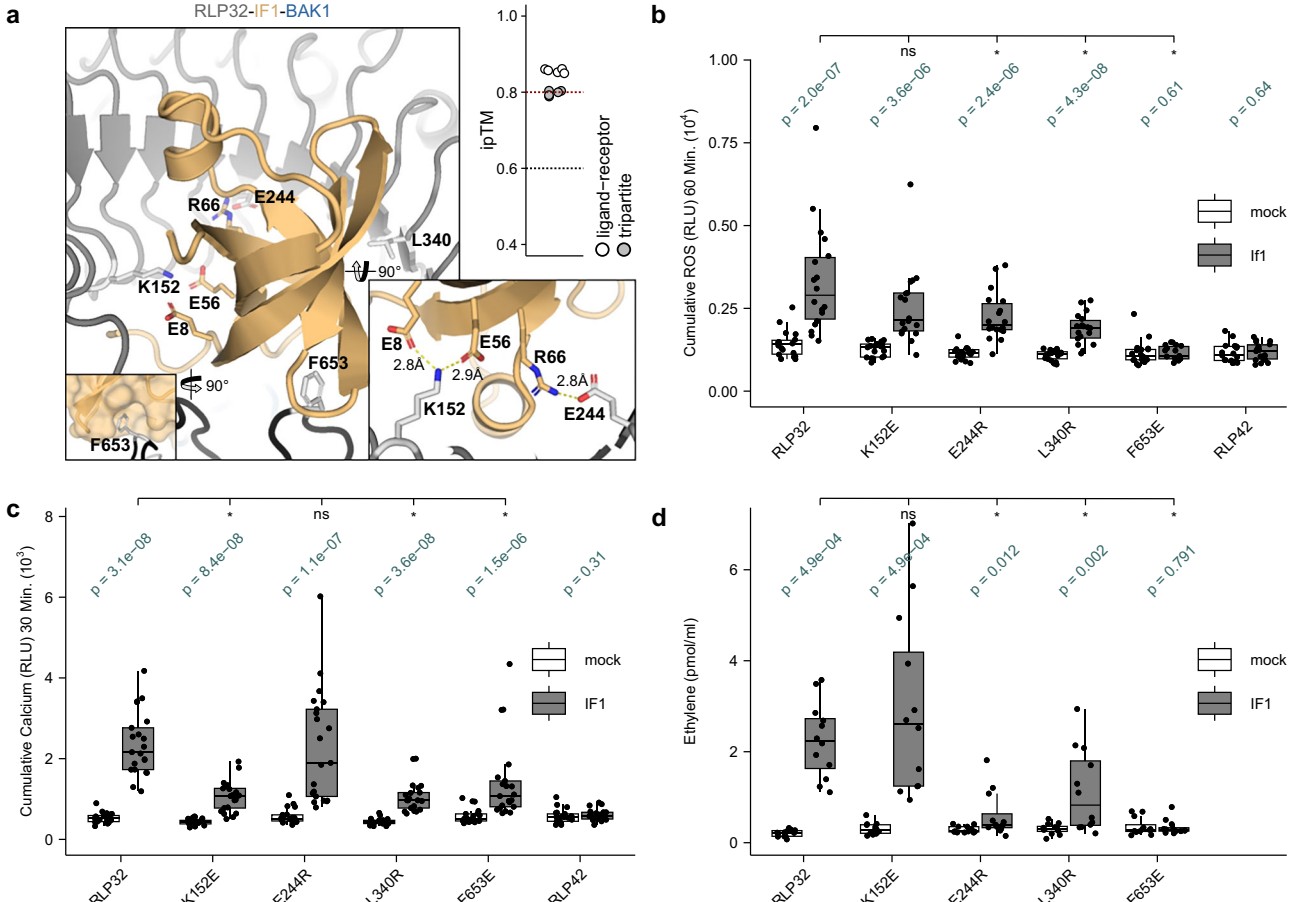

**Fig. 3 | Single AA changes within the predicted ligand-binding interfaces affect the functionality of RLP32. a** AF3 predicts a high ipTM for RLP32 in complex with IF1 and the co-receptor BAK1. The AF3 guidelines state that ipTM values higher than 0.8 represent confident high-quality predictions. ipTM values between 0.6 and 0.8 are within a gray zone where predictions could be correct or incorrect. The AF3 cutoffs are depicted with a red and black dotted line. pLDDT scores can be found in Supplementary Fig. 2c, f. The ID is highlighted in black, other LRR-RP domains in dark gray. The ligand is depicted in yellow and BAK1 in blue. Residues highlighted in light gray were mutagenized from the receptor side in this study. **b**–**d** Diverse responses in *N. benthamiana* post heterologous expression of RLP32 and RLP32 variants and subsequent treatment with $H_2O$ (white) or 1 μM IF1 (gray). Each biological replicate is represented by at least three technical replicates. Box plots indicate the median and the interquartile range (IQR), the whiskers extend to the

most extreme data points within 1.5 times the IQR. Significance was tested by performing non-parametric two-sided Wilcoxon-Mann-Whitney tests between both mock- and ligand-treated RLP32 (variants) (depicted in green), as well as the ligand-treated RLP32 vs specific RLP32 variants, without adjustments for multiple comparisons. The asterisks indicate a significant difference of $p < 0.05$. **b** ROS production (4-60 min) in cumulative RLUs post treatment. Eight independent biological replicates ($n = 8$ plants) were performed. **c** Cytosolic $Ca^{2+}$ concentrations (3–30 min) in cumulative RLUs post treatment. Seven independent biological replicates ($n = 7$ plants) were performed. **d** Ethylene accumulation after 4 h treatment. Data points are indicated as gray dots from four independent experiments. The Pdb file can be found in Supplementary Data 1 and the predicted local distance difference test (pLDDT) scores can be found in Supplementary Fig. 2c, f, i.

## Conserved residues within the CT part of the ID are crucial for BAK1 recruitment and LRR-RP function

To understand how LRR-RPs recruits BAK1 for immune activation, we analyzed the tripartite predictions of the ligand-bound ectodomains of RLP23, RLP42, and RLP32 with the ectodomain of BAK1. The predictions show that the CT part of the IDs of RLP23, RLP42, and RLP32 adopts an antiparallel β-sheet (Fig. 4a). Upstream of the CT part, the conserved tyrosine residue in the Y-$x_8$-KG motif of LRR-RP (RLP23[Y678], RLP42[Y680], and RLP32[Y656]) located at the end NT part of the ID likely guides the position of the CT part of the ID. The lysine residue in the Y-$x_8$-KG motif interacts with the main chain of the BAK1 N-terminal capping domain. At the second position of the penultimate LRR motif, RLP23, RLP42 and RLP32 have a glutamate residue that forms an ionic interaction with the lysine residue and cooperatively interact with the main chain of the BAK1 N-terminal capping domain (Fig. 4a).

To test the putative mechanistic role of this interface, mutant variants were created with single AA substitutions of the conserved tyrosine, lysine and glutamate residues of each LRR-RP. In addition,

RLP23[Y688A] was created, in which we mutated a non-conserved residue relative to RXEG1 (W806A) crucial for BAK1 recruitment by RXEG1 upon XEG1 binding[6].

Notably, RLP42[E751A] and RLP32[E727A] abolished all measured immune responses, whereas RLP23[E751A] significantly reduced ROS and $Ca^{2+}$ bursts (Fig. 4b–d). In addition, the RLP42[K689A] and RLP32[K665A] mutations consistently reduced or abolished all measured immune responses while RLP23[K689A] significantly reduced ROS and ethylene productions (Fig. 4b–d). Finally, RLP23[Y678A] and RLP32[Y656A] consistently reduced or abolished all measured immune responses, while RLP42[Y680A] significantly reduced ROS and ethylene productions. Importantly, all LRR-RP variants of the conserved residues had reduced complex formation with BAK1 (Fig. 4e).

In summary, these results functionally validate (1) the importance of the glutamate residue at the second position of the penultimate LRR across different LRR-RPs, (2) the importance of the lysine residue in the linking loop of the LRR-RP IDs, and (3) the novel characterized role of the tyrosine residue within the ID of LRR-RPs with the Y-$x_8$-KG motif.

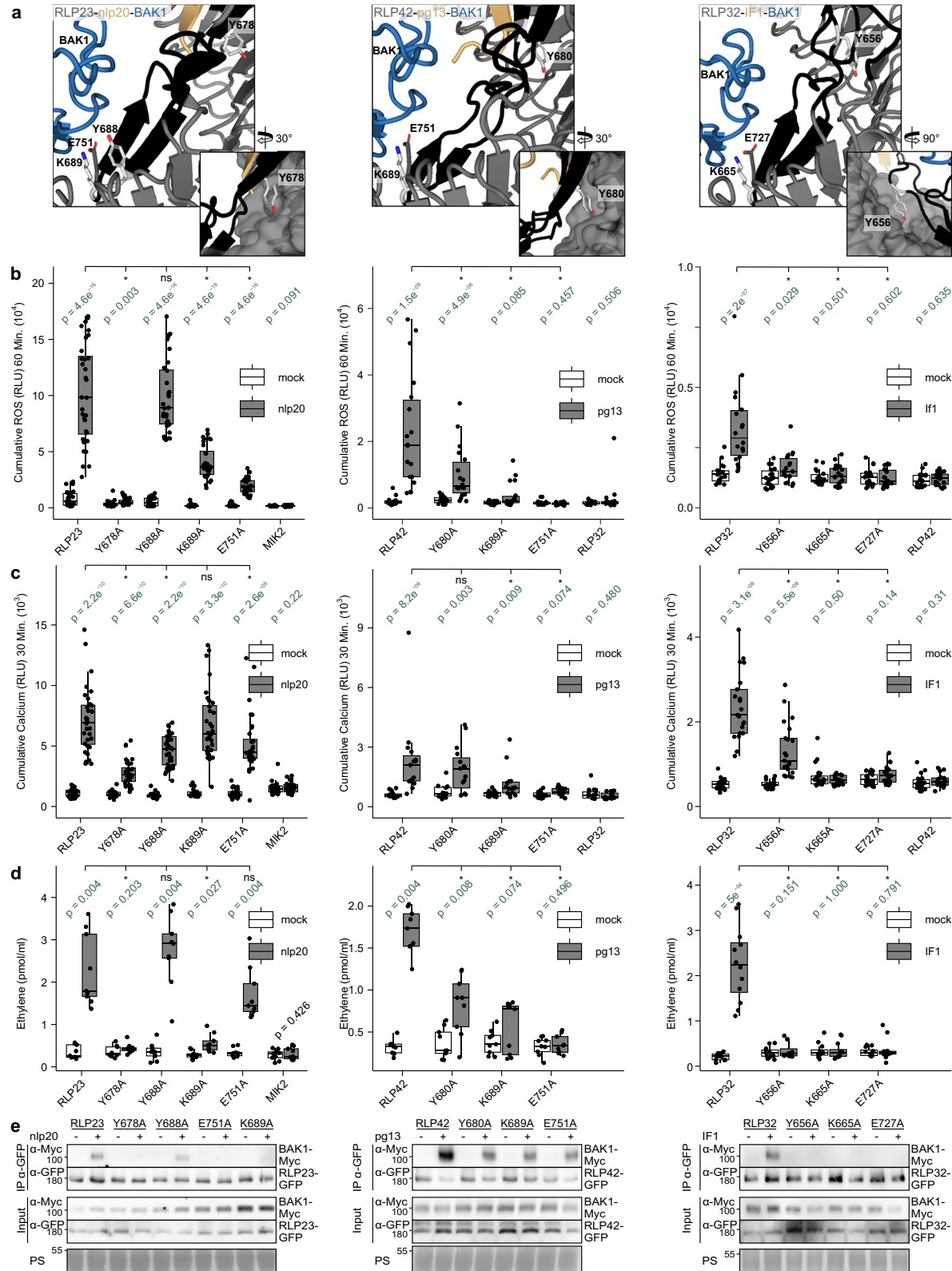

## Discussion

LRR-RP-type PRRs are of interest for augmenting crop disease resistance through engineering[18]. We set out to extend our proof-of-concept of AI-based structural modeling beyond just the ligand-binding interface of the structural more simplistic LRR-RKs with a short linear peptide[21]. We present a rapid and inexpensive approach for resolving LRR-RP receptor complex interfaces, now including the co-receptor interface and diverse ligands in length and tertiary structure. This provides an alternative to classical structural approaches for LRR-RPs which are allegedly challenging to accomplish, with RXEG1 being so far the sole structurally resolved LRR-RP[6]. By validating three AI-predicted LRR-RP receptor complexes, we now shed light on the diversity of ligand-binding mechanisms across LRR-RPs, while

**Fig. 4 | A conserved tyrosine within the NT part of the ID, predicted to stabilize the ID domain upon ligand binding, is critical for RLP23 (left), RLP42 (middle) and RLP32 (right) function. a** Structural representations of the tripartite complexes of RLP23-nlp20-BAK1, RLP42-pg13-BAK1 and RLP32-IF1-BAK respectively from left to right. pLDDT scores can be found in Supplementary Fig. 2g-i. The ID is highlighted in black, other LRR-RP domains in dark gray. The ligand is depicted in yellow and BAK1 in blue. Residues highlighted in light gray were mutagenized in this study. Besides the conserved Y of interest within the ID, this includes a conserved K within the ID and E within the CT LRR earlier characterized as BAK1 interacting residues for RXEG1. **b–d** Diverse responses in *N. benthamiana* post heterologous expression of the respective RLPs and their variants and subsequent treatment with $H_2O$ (white) or 1 µM of the respective ligand (gray). Each biological replicate is represented by at least three technical replicates. Box plots indicate the median and the interquartile range (IQR), the whiskers extend to the most extreme data points within 1.5 times the IQR. Significance was tested by performing non-parametric two-sided Wilcoxon-Mann-Whitney tests between both mock and ligand for each receptor (variant) (depicted in green), as well as the ligand-treated WT receptor vs specific variants, without adjustments for multiple comparisons. The asterisks indicate a significant difference of $p < 0.05$. **b** Shown is ROS production (4-60 min) in cumulative RLUs post treatment with $H_2O$ (white) or the respective ligand (1 µM IF1, gray). At least six independent biological replicates ($n = 8$ (RLP23), or 6 (RLP42 and RLP32) plants) were performed. **c** Shown are increases in $Ca^{2+}$ cytosolic concentrations (3–30 min), in cumulative RLUs post treatment with $H_2O$ (white) or the corresponding peptides. At least five independent biological replicates ($n = 8$ (RLP23), or 5 (RLP42), or 7 (RLP32) plants) were performed. **d** Ethylene accumulation after 4 h treatment. Data points are indicated as gray dots from at least three independent experiments ($n = 3$ (RLP23, RLP42) or 4 (RLP32) plants). **e** Proteins extracted from *N. benthamiana* leaves expressing the respective GFP tagged LRR-RP in combination with Myc-tagged BAK1 and treated with water (−) or 1 µM elicitor (+) for 5 min before collecting were used for co-immunoprecipitation with GFP-trap beads and immunoblotting with tag-specific antibodies. IP, immunoprecipitation PS, Ponceau S. The Pdb files can be found in Supplementary Data 1 and the predicted local distance difference test (pLDDT) scores can be found in Supplementary Fig. 2c, f, i.

additionally revealing a conserved role for the ID in co-receptor recruitment.

AI-based predictions overcome the bottleneck of producing purified proteins, which appear to be challenging for LRR-RPs and extracellular domains/proteins in general[30]. However, the success rate of complex predictions remains a challenge relative to monomer predictions[31]. pLDDt and ipTM are initial indicators of prediction quality but beware of false positives and negatives, functional validation is ultimately essential to leverage the full potential of AI-based structure predictions[18,21,30,32,33]. To gain trust in such predictions, before proceeding with time-consuming and costly experimental validation of receptor complexes, comparative genomics as well as reevaluating existing variants on both sides of the interaction can be informative, as demonstrated in this study and earlier for the MIK2-SCOOP interaction[18,21].

Considering RLP23 and RLP42, earlier published characterizations and alanine-scanning mutagenesis approaches of NLPs and PGs were used to gain trust in the predicted receptor complexes and help design receptor variants for experimental validation[9,26]. In contrast, the characterization of IF1 did not provide critical residues for RLP32 sensing but instead revealed the importance of the IF1 tertiary structure[28]. Supportive of the complex prediction of RLP32-IF1-BAK1 receptor complex, IF1 appears in a highly similar conformation bound to 30S ribosomal subunit[29]. Structural studies of IF1 also support why mutating AAs in the a-helix of IF1 was not sufficient to reduce IF1 elicitor activity, as the loop is known for having structural flexibility and binds 16S rRNA in the ribosome initiation complex[28,34,35].

Finally, from the receptor side of the interaction: (1) earlier identified residues impacting RLP42 function were reevaluated, and supported the AF3 prediction of the RLP42-pg13-BAK1 receptor complex, (2) RLP32 residues at the interaction interface are conserved within putative RLP32 orthologues earlier identified through synteny[18], and (3) conserved residues of importance for BAK1 recruitment by RXEG1 upon ligand binding were predicted to function similarly for RLP23, RLP42 and RLP32[6]. Ultimately, we functionally validated the three predicted LRR-RLP receptor complexes by testing receptor variants across multiple interaction interfaces through quantification of a multitude of plant immune signaling outputs, such as ROS production, $Ca^{2+}$ burst, ethylene production, and ligand-induced complex formation with BAK1 (Figs. 2–4).

Intriguingly, the NT part of the IDs of RLP23 and RLP42 adopt an antiparallel β-sheet conformation, and ligand-binding is mainly mediated through a β-strand interaction with their respective ligands nlp20 (20 AAs) and pg13 (13 AAs) besides interaction with the LRR domain. The NT part of the RXEG1 ID also adopts an antiparallel β-sheet conformation but the characterized XEG1- interacting AA residues are located within the linking loop. Moreover, the RXEG1 NT loopout is also involved in XEG1-binding whereas RLP23 and RLP42 lack NT loopouts[4,6]. Hence, the shared ligand-binding mechanism of RLP23 and RLP42 is diverse compared to RXEG1-XEG1-BAK1. Intriguingly, the LRR-RLK-Xb PHYTOSULFOKINE RECEPTOR 1 (PSKR1) also harbors a two-stranded antiparallel β-sheet in its ID and binds the disulfated penta-peptide ligand phytosulfokine (PSK) mainly through forming an anti β-sheet interaction with the β-strand from the ID[36]. In contrast, the brassinosteroid receptor BRASSINOSTEROID INSENSITIVE 1 (BRI1) ID contains a three-stranded antiparallel β-sheet and forms a binding pocket for the steroid hormone together with the BRI1LRR core (LRR 21-25)[37]. Besides nlp20 and pg13, other relatively small characterized LRR-RP ligands are inceptin (11 AAs, In11) and Avr9 (28 AAs) sensed by cowpea INR and tomato Cf-9, respectively[5,38–40]. However, In11 and Avr9 are non-linear peptides that contain respectively one and three disulfide bridges, which at least for Avr9 is required for eliciting activity, indicating the importance of its tertiary structure for Cf-9 sensing similar to IF1 and XEG1[39]. Moreover, both INR and Cf-9 LRR domains contain long NT loopouts, suggesting diverse ligand-binding mechanisms relative to RLP23 and RLP42[4].

The minimal active IF1 ligand (residues 7-72) is relatively large, and the importance of its tertiary structure for RLP32 sensing has been suggested[6,28]. We therefore focused on the putative interaction of RLP32 with the five-stranded β-barrel fold of IF1 as (1) a triple mutant (K39L R41L K42L) in the α-helical motif of IF1 that showed flexibility in the NMR structure was not sufficient to reduce IF1 elicitor activity, and (2) the relatively short eight-AA RLP32 NT loopout positioned in the NT LRR capping domain was predicted with lower confidence[28,35]. The IF1 OB fold forms an extensive interaction interface with LRR domain and the NT part of ID domain of RLP32, mediated by electrostatic and hydrophobic interactions. The NT part of the ID domain of RLP32 adopts a loopout shape with F653 buried inside the hydrophobic pocket of IF1. This contrasts to RLP23, RLP42 and RXEG1 as the NT part of the ID does not adopt an antiparallel β-sheet. Like RXEG1 binding the active conformation of XEG1, our observations indicate that RLP32 evolved to perceive IF1 in its active state by interacting with its structurally more conserved region, – a preferred template for PRR evolution –, the five-stranded β-barrel fold[6].

Overall, our study revealed the ligand-binding mechanism of three LRR-RPs. In doing so, we shed light on the diversity of LRR-RP binding mechanisms with a consistent prominent role for the NT part of the ID in ligand-binding, although through diverse mechanisms.

The ligand-induced association with SERK co-receptors such as BAK1 has been characterized for many LRR-RPs[4,10,28,38,41–43]. Through the generation of RLP23, RLP42 and RLP32 variants, we now demonstrate the conserved functional role for the CT part of the ID of LRR-RPs in BAK1 recruitment. The N-terminal capping region of BAK1(LRR) interacts with a binding groove created between the CT LRRs and the

CT part of the ID which adopts an antiparallel β-sheet conformation. The lysine residue, present in both the Y-x$_8$-KG and K-x$_5$-Y motifs, interacts with the main chain of the BAK1 N-terminal capping domain. At the second position of the penultimate LRR motif, a glutamate residue forms ionic interaction with the lysine residue and cooperatively interacts with the main chain of the BAK1 N-terminal capping domain (Fig. 4a). The lysine is positioned within the linking loop of the two parallel strands forming the CT ID β-sheet of LRR-RPs of the structural predictions of the RLP23, RLP42 and RLP32 receptor complex and the structure of the RXEG1 receptor complex (Fig. 4a)[6]. Across all four LRR-RPs, a single AA change to alanine was sufficient to affect BAK1 interaction (Fig. 4)[6]. In silico analysis indicates strong conservation of the lysine residue within the CT part of the ID across LRR-RPs. Indeed, 77 % of 14'315 LRR-RPs identified in a 350-genome dataset covering the plant kingdom contain either the K-x$_5$-Y or Y-x$_8$-KG motif, whereas the glutamate residue at the second position of the penultimate LRR motif is conserved in 86 % of them[4,8]. Importantly, less than 4 % of the IDs of subfamily-Xb LRR-RKs harbor either the K-x$_5$-Y or the Y-x$_8$-KG motif. Structural comparisons of RLP23, RLP42, RLP32 and RXEG1 receptor complexes relative to the LRR-RK-Xb receptor complexes of BRI1 and PSKR1 also indicate different roles of the respective IDs in co-receptor recruitment but a similar role of the C3 regions of LRR-RPs and subfamily-Xb LRR-RKs[6,8,36,44,45]. Intriguingly, the uncharacterized tyrosine residue of Y-x$_8$-KG in RLP23, RLP42 and RLP32 is consistently in the proximity of the LRR backbone and single AA changes to alanine affect LRR-RP function and BAK1 interaction. The Y-x$_8$-KG is conserved in 43 % of LRR-RPs. Physiological data of the respective LRR-RP variants suggests the importance for correct positioning of the CT part of the ID through the conserved tyrosine residue (Fig. 4). Moreover, within RLP23 and RLP42, this tyrosine residue is positioned at the end of the NT antiparallel β-sheet at the ligand-binding interface, and might thus also be crucial for stabilizing the NT part of the ID. Hence, our data confirms a conserved mechanistic role of the ID for co-receptor recruitment conserved across LRR-RPs besides the importance of earlier characterized residues in the CT LRRs of subfamily-Xb LRR-RKs and LRR-RPs. More specifically, our structural analysis of LRR-RPs RLP23, RLP42 and RLP32 and the earlier published structure of RXEG1 highlight the importance of the BAK1-binding groove formed by the LRR domain and the CT part of the ID through the formation of an antiparallel β-sheet.

Our results open avenues for a molecular understanding of the co-evolution between an important class of plant immune receptors represented by LRR-RPs and plant pathogens, as well as the engineering of such receptors for improved crop disease resistance.

## Methods

### AlphaFold 3 (AF3)

AF3 protein structure complex predictions of the extracellular domains of RLP23 (AT2G32680), RLP32 (AT3G05650) and RLP42 (AT3G25020), the corresponding ligands, with or without the extracellular domain of BAK1 (AT4G33430) were created using AlphaFold Server Beta[25,46,47]. The extracellular domain of the LRR-RPs and BAK1 were determined using deepTMHMM as irrelevant domains might obstruct putative interactions[21,33,48]. The structure files (.pdb) of the successfully predicted complexes for predictions are provided in Supplementary Data 1. Predicted models obtained from AF3 were visualized and structurally analyzed using PDBePISA and PyMOL[49,50], retrieved from (http://www.pymol.org/pymol). R and the R-packages dplyr (v1.1.2), ggpubr (v.0.6.0), and ggplot2 (v3.4.2) were used to analyze and plot the ipTM data. The resulting figures were edited in Corel-DRAW Home & Student x7.

### Molecular cloning and mutagenesis

All primers and plasmids used and generated in this study are listed (Supplementary Table 1, Supplementary Data 2). Considering RLP23 variants, site-directed mutagenesis (SDM) was conducted as earlier described[51]. The GoldenGate L0 construct of Arabidopsis *RLP23* was used as template (CZLp1779). The PCR reaction was digested with DpnI (New England Biolabs) at 37 °C for 2 h without prior clean-ups and then transformed into *E. coli* DH10b. Subsequently, the L0 *RLP23* fragments and an mEGFP C-terminal tag (CZLp4772) were inserted into the Golden-Gate L1 plasmid CZLp4130, which already includes a 35S promotor (CaMV) and an OCS terminator. GoldenGate reactions were performed with 5 U of restriction enzyme, 200 U of T4 ligase in T4 ligase buffer (NEB), 0.1 mg/mL BSA (NEB) and 40 GoldenGate digestion ligation cycles[52]. All constructs were validated by whole plasmid sequencing (Eurofins genomics). *RLP32* wild-type and variant constructs were generated using Gibson assembly Master Mix (NEB) into the SpeI site of pLOCG, with 35S promoter and C-terminal GFP fusion. *RLP42* variant constructs were generated by Genscript and similarly subcloned into the SpeI site of pLOCG, with 35S promoter and C-terminal GFP fusion.

### Transient expression in *Nicotiana benthamiana*

*N. benthamiana* does not respond to nlp20, pg13 and IF1, allowing the use of heterologous expression in *N. benthamiana* to test the putative function of the corresponding receptors RLP23, RLP42 and RLP32[9,10,28]. *Agrobacterium tumefaciens* strain GV3101 transformed with the appropriate construct were grown overnight in LB-media and spundown. The bacteria were resuspended in infiltration media (10 mM MES-KOH, pH 5.8, 10 mM MgCl$_2$) and adjusted to an OD$_{600}$ of 0.5. After 3 h of incubation, the youngest fully expanded leaves of 4- to 5-week-old plants were infiltrated.

### Protein extraction and western blotting

*N. benthamiana* leaf tissues were flash-frozen in liquid nitrogen and ground using plastic pestles in 1.5-mL microcentrifuge tubes. Ground tissue was mixed with 2× loading sample buffer (4 % SDS, 20 % glycerol, 20 mM DTT, 0.004 % bromophenol blue, and 100 mM Tris-HCl pH 7.5) for 10 min at 95 °C. Subsequently, samples were spun at 13,000 × g for 2 min prior to loading and run on 1.5-mm 10 % SDS-PAGE gels. Proteins were transferred onto a PVDF membrane (ThermoFisher) prior to incubation with α-GFP (B-2) HRP (Santa Cruz 9996 HRP, 1:1500). Western blots were imaged with a Bio-Rad ChemiDoc or e-Blot Touch Imager and Image Lab Touch Software (v2.2.0.08) and eBLOT14, respectively. Protein loading was visualized by staining the blotted membrane with Coomassie brilliant blue (CBB).

### Confocal imaging

Protein localization in *N. benthamiana* was analyzed two-three days after transient infiltration with *A. tumefaciens* strains expressing different RLP32 and RLP42 variants tagged with C-terminal GFP. Fluorescence detection was performed in a Leica Stellaris with wavelength emission at 488 nm and detection at 494-560 nm for GFP and 625-750 nm range for chlorophyll autofluorescence. Images were extracted with LAS X using consistent parameters for the GFP channel.

### Peptides

All peptides used in this study were synthesized and firstly resuspended in H$_2$O (pg13 and flg22) or DMSO (nlp20 and IF1) to generate 10 mM stock solutions. Nlp20 (AIMYSWYFPKDSPVTGLGHR), pg13 (AAHNSDGFDVSSS) and IF1 (MAKEDNIEMQGTVLETLPNTMFRVE-LENGHVVTAHISGKMRKNYIRILTGDKVTVELTPYDLSKGRIVFRSR) were used to test the activation of the corresponding receptors RLP23, RLP42 and RLP32[9,10,28]. The flg22 peptide (QRLSTGSRINSAKDDAAGL-QIA) originates from bacterial flagellin and was used as a positive control for ROS production and cytoplasmic calcium measurements upon peptide treatment[53].

### ROS measurements in *Nicotiana benthamiana*

Following Agrobacterium infiltration for receptor expression (48 h), leaf punches were taken with a 4-mm biopsy punch and floated in 100 µL of H$_2$O using individual cells of a white 96-well bottom plate (Greiner F-Boden, lumitrac, med. Binding, [REF 655075]). After overnight incubation, H$_2$O was removed and ROS production was measured upon addition of a 100-µL assay solution which contained 10 µg/mL horseradish peroxidase (P6782, Merck), 10 mM luminol and the treatment (1 µM ligand of interest, 0.1 µM flg22 or H$_2$O). Luminescence was quantified with a HIGH-RESOLUTION PHOTON COUNTING SYSTEM (HRPCS218, Photek). Biological replicates were quantified (n = 8 plants), with each biological replicate representing four technical replicates. R and the R-packages dplyr (v1.1.2), ggpubr (v.0.6.0), and ggplot2 (v3.4.2) were used to analyze and plot the data. The resulting figure was edited in Corel-DRAW Home & Student x7.

### Cytoplasmic calcium measurements in *Nicotiana benthamiana*

Following Agrobacterium infiltration for receptor expression (24 h) in a stable aequorin expressing line of *N. benthamiana*[54], leaf punches were taken with a 4-mm biopsy punch and floated in 100 µL of H$_2$O with 20 µM coelenterazine (Merck), using individual cells of a white 96-well bottom plate (Greiner F-Boden, lumitrac, med. Binding, [REF 655075]). After overnight incubation, the coelenterazine solution was replaced with 100 µL H$_2$O and rested for circa 30 min in the dark. Luminescence was quantified in a Berthold Tristar 3 plate reader every minute for 5 min before and 30 min post elicitor treatment using an integration time of 250 ms. Biological replicates were quantified (n ≥ 5 plants), with each biological replicate representing four technical replicates. R and the R-packages dplyr (v1.1.2), ggpubr (v.0.6.0), and ggplot2 (v3.4.2) were used to analyze and plot the data. The resulting figure was edited in Corel-DRAW Home & Student x7.

### Ethylene measurements

After 48 h Agrobacterium infiltration, leaves were cut into pieces (about 0.5 × 0.5 cm) and floated on water overnight. Three leaf pieces were incubated in a sealed 6.5-mL glass tube with 0.4 mL 20 mM MES buffer pH 5.7 and the indicated elicitors. Ethylene production was measured by gas chromatographic analysis (GC-14A, Shimadzu) of 1 mL air from the closed tube after incubation for 4 h.

### Co-immunoprecipitations

For immunoprecipitations, membrane proteins of Agrobacterium-infiltrated *N. benthamiana* leaves were extracted at 1 mg ml$^{-1}$ in extraction buffer [50 mM Tris-HCl, pH 7.4, 150 mM NaCl, 0.25% deoxycholic acid, 1 % NP-40, 1 mM EDTA, proteinase inhibitor cocktail (Roche)] and immuno-adsorbed by means of their GFP tags on GFP-trap agarose beads (ChromoTek)[41]. Immunoblots were developed either directly with anti-GFP antibodies (Torrey Pines Biolabs TP401, 1:10,000 dilution) or anti-myc antibodies (Sigma-Aldrich C3956, 1:10,000 dilution), followed by HRP conjugated anti-Rabbit (Agrisera AS09 602, 1:20,000). Chemiluminescence was detected with the ECL Western blotting detection system (GE Healthcare) and a CCD camera (Amersham Imager 600).

### Reporting summary

Further information on research design is available in the Nature Portfolio Reporting Summary linked to this article.

## Data availability

The authors declare that the data supporting the findings of this study are available within the manuscript, supplementary files and source data. The source data includes all plotted data, predicted receptor complexes, plasmid maps, original western blot images and original co-IP images. Source data are provided with this paper.

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

## Acknowledgements

This research was supported by the University of Zurich (CZ), an UZH Postdoc Grant (K-74503-08-01, SS), an SNSF Postdoctoral Fellowship (TMPFP3_224980, GK), an EMBO Postdoctoral Fellowship (ALTF 580-2022, AFF) and the German Research foundation (CRC1101-D10, Nu70/19-1, TRR356-B05, TN). We thank all members of the Zipfel lab for discussions. We particularly thank Neftaly Cruz Mireles, Harshith CY, Marie Le Naour--Vernet, Dennis Mahr and Keran Zhai for their feedback on the manuscript.

## Author contributions

S.S. and C.Z. conceptualized and designed the experiment; T.N. and C.Z. supervised and acquired funding; L.Z., S.S., V.S., and A.D.F.F. conducted the experiments; S.S. and L.Z. performed data analysis, S.S. performed the bioinformatics and visualizations, S.S. and G.K. interpreted and visualized the predicted structures; S.S. wrote the paper; S.S., G.K. and C.Z. reviewed and edited the manuscript; and all other authors commented and agreed on the manuscript.

## Competing interests

The authors declare no competing interest.
