## [Transparent Peer Review file · Nature Communications]

β -sheet stabilization of the island domain underlies ligand-induced LRR-RP activation of plant immune signaling

Corresponding Author: Professor Cyril Zipfel

Version 0:

Reviewer comments:

Reviewer #1

(Remarks to the Author)

In this manuscript, Snoeck et al utilise alphafold modelling to predict the binding mechanism of ligands to three different LRR-RPs. LRR-RPs represent a large family of plant immunity receptors capable of recognising peptides and proteins from plant pathogens. The focus of this study was RLP23 and 42 which recognised small peptides and RLP32 which recognises the protein IF1.

The authors demonstrate that AF3 can generate quality binding mode predictions. They validate these predictions via mutagenesis of both the ligand and receptor using the analysis of downstream ROS, calcium and ethylene quantification. These responses are known outputs of LRR-RP activation, and the inclusion of all assays adds robustness to the study and the authors conclusions. In some instances, co-IPs are used to validate that the loss of binding impacts the interaction between the LRR-RP and BAK1.

Generally, we agree with the conclusions that the authors make based on the data presented. The manuscript itself is easy to follow with a logical structure and in general well presented figures (see comments below).

While we find that the work as presented support the models and findings, the authors are ultimately using a limited data set to largely validate a mechanism that was described in the experimental structure of RXEG1 – XEG1 – BAK1. The novelty here is the approach (although using alphafold for similar studies is now common practice) and the observation that peptide and/or protein binding likely cause LRR-RP to engage BAK1 in the same way. We don't wish to sound critical of the work itself, we like the paper, the data appears robust and the work demonstrates that the approach used is useful for the study and engineering of these receptors, but it is our opinion that the novelty and advance provided by the study are a little overstated. Toning this down in a revised version would be advised.

Comments:

1. Line 20, I'm not sure how 'novel' this ligand-binding mechanism really is, given that it has been uncovered structurally before. Conserved would be a better choice of words in this case.
2. Line 81 and 84, replace the word with experimental rather than traditional and classical
3. Line 122, it is not clear if the authors are talking about backbone interactions with the glycine residues that are mediate the antiparallel beta-sheet interactions? Have they considered passing their models through something like PISA to show the actual interactions involved in these beta-sheet interactions and thereby ligand binding? It would be nice to see some H-bonds and rough buried surface area calculations to support these observations.
 - Related to this is the statement in line 143 indicating that their approach to mutate G677, G699 was to destabilise interactions with Y4 and W6 – would be good to give the reader an idea of how these residues stabilise the interaction in the first place as it is not obvious from figure 1 up until this point in the manuscript.
4. Figure 1, color scheme is poor. We would recommend making the ligand yellow throughout and simply highlighting the residues of interest. We realise this is not possible for the receptor due to the involvement of the glycine so color changes make sense here.
5. Section 175-189, these are not results. This section seeks to provide a structural context (based on the models presented here) to previously identified and studied mutations. Ultimately it is and interpretation based on a structural model, with no additional data added. It does not belong in the results. In many ways this could all be summarised in supplementary data as

a table and perhaps some discussion dedicated to this.

6. Line 301, use of the word architectural appears to be out of context

7. Line 377, the statement is a little strong given this mutation in RLP42 can still recruit BAK1 (Fig. 4E)

8. The island domains (IDs) of the LRR-RLPs and the subfamily Xb-LRR-RKs are different in terms of the presence of the lysine-containing motifs as per the authors, thereby having different roles in BAK1 recruitment. Has there been any work done to show how different orthologues of the BAK1 co-receptors are recruited by these two classes of LRRs? If not, would this be an angle worth exploring for the authors in support of one of their discussion points in line 316 on 'comparative genomics and reevaluating existing variants on both sides of the interaction'

Reviewer #2

(Remarks to the Author)

Reviewer #3

(Remarks to the Author)

This paper seeks to use an AI/computational approach (AlphaFold) to better understand how plant immune receptors bind to their ligands. Here the authors find that several members of the RLP subclass of LRR receptors (lacking an intracellular kinase domain) have a non-LRR "island domain" embedded within the LRR, and can "loop out" of the typical LRR horseshoe fold. This domain has been previously shown to be involved in substrate binding by x-ray crystallography. In three different RLP-ligand systems, AlphaFold satisfyingly predicted the ID was involved in ligand binding, and also coordinated the interaction of the RLP/ligand complex with BAK1, the kinase protein, required for downstream signaling. These hypothetical structures were then validated by mutagenizing residues (often conserved) that were located in key interaction surfaces. In some cases the mutations were pre-existing in the literature, and the structures allowed for the opposing residues to also be probed for function. The authors went on to test both a variety of downstream signaling outputs, as well as assessing interaction via co-immunoprecipitation. In general the loss of interaction via coIP correlated well with loss of signaling assuming that we can invoke different thresholds for a loss-of-function in the various assays (i.e. how much loss of interaction is expected to result in a loss of signaling? – hard to say).

In general, the paper is clear and well-written. I appreciated that the authors clearly stated that these models were hypotheses to be tested (and then did so). It would have been nice to see the authors really nail down the reality of what is going on by making compensatory receptor/ligand mutations that are loss-of-function alleles individually, but can function when combined together. But there is no guarantee that this approach is always possible, and I'm satisfied with the evidence presented. It is nice to see the authors include the two related ligands, and also a second ligand class, to show that the approach is at least somewhat generalizable.

This story should be of wide interest to the plant immunity field, and potentially beyond to other LRR-containing proteins across the tree of life.

There are a few minor points and graphical changes that could improve readability that I will list below. I don't have any major concerns, nice story.

Individual points:

It would be extremely useful to renumber the amino acid sequences of the PDB files so that they match the figures. If not, every reader has to go dig up the full-length RLP sequences and do it themselves to try to reconcile them with the figures.

Figures: I think that the main figures would benefit from some representation of where we are on the complex. This "zoomed-out" big picture is indeed in the supplemental, but it'd be really useful for the reader to understand this without digging. If no space, fine.

Figure 3A: I found this figure not that helpful on its own for understanding the modeled complex. The main text is also vague on which residues are actually interacting, how far they are apart, etc. I'd consider color-coding the important residues by protein. Would be nice to have a zoom-in on the proposed ionic interactions to show how they are oriented and that the distances are reasonable. I don't think that the space-filling overlay is useful here, and it adds clutter to an already complex image (also its not used in the other figures)

Fig 3 B-D; The presentation of statistics is a bit confusing (* for some, and p-value for others even though the test is the same?). If you could color-code these and/or add more explanatory text to the legend it would be much more reader friendly.

Figure 4: I think that this figure is not making clear what the tyrosine is doing. You could add a zoom in that shows the tyrosine relative to the opposing beta-sheets, or similar to clarify. As is, I get a better understanding of the charged residues, while the title of the figure focuses on the tyrosine. Similarly to 3A, you could color-code residues to make it more obvious what was going on (e.g. would be more clear that K665 is on the black ID loop, and E727 is on the LRR).

Figure 4A: K655 is typo, should be K665.

CoIPs: An empty vector control on the input blots to show that the bands are indeed what we expect them to be would have been nice.

Version 1:

Reviewer comments:

Reviewer #1

(Remarks to the Author)

The authors have addressed our comments and any concerns raised during initial review.

Reviewer #3

(Remarks to the Author)

The authors have done a good job of addressing my concerns.

Dear reviewers,

Thank you very much for the evaluation of our manuscript.

We have done our best to address the comments, and we provide below a point-by-point response. We hope that our revision will be satisfactory and that our manuscript can be accepted for publication in Nature communications.

Best wishes,

Cyril Zipfel and Simon Snoeck, on behalf of all authors

Response to Reviewer 1

In this manuscript, Snoeck et al utilise alphafold modelling to predict the binding mechanism of ligands to three different LRR-RPs. LRR-RPs represent a large family of plant immunity receptors capable of recognising peptides and proteins from plant pathogens. The focus of this study was RLP23 and 42 which recognised small peptides and RLP32 which recognises the protein IF1.

The authors demonstrate that AF3 can generate quality binding mode predictions. They validate these predictions via mutagenesis of both the ligand and receptor using the analysis of downstream ROS, calcium and ethylene quantification. These responses are known outputs of LRR-RP activation, and the inclusion of all assays adds robustness to the study and the authors conclusions. In some instances, co-IPs are used to validate that the loss of binding impacts the interaction between the LRR-RP and BAK1.

Generally, we agree with the conclusions that the authors make based on the data presented. The manuscript itself is easy to follow with a logical structure and in general well presented figures (see comments below).

While we find that the work as presented support the models and findings, the authors are ultimately using a limited data set to largely validate a mechanism that was described in the experimental structure of RXEG1 – XEG1 – BAK1. The novelty here is the approach (although using alphafold for similar studies is now common practice) and the observation that peptide and/or protein binding likely cause LRR-RP to engage BAK1 in the same way. We don't wish to sound critical of the work itself, we like the paper, the data appears robust and the work demonstrates that the approach used is useful for the study and engineering of these receptors, but it is our opinion that the novelty and advance provided by the study are a little overstated. Toning this down in a revised version would be advised.

Our response:

We thank the reviewer for the positive feedback and appreciation of our work. We address the raised comments point-by-point below.

Comments:

1. Line 20, I'm not sure how 'novel' this ligand-binding mechanism really is, given that it has been uncovered structurally before. Conserved would be a better choice of words in this case.

Our response: We thank the reviewer for this suggestion. Relative to RXEG1 (the only LRR-RP structure available), this would be novel. However, we agree that this perspective might be

confusing as the revealed binding mechanism shows similarities with the PSK-PSKR binding mechanism from a diverse receptor family, the LRR-RLK-Xb family, as discussed (line 342). Therefore, we removed “novel” consistently across the manuscript.

2. Line 81 and 84, replace the word with experimental rather than traditional and classical

Our response: We thank the reviewer for this suggestion. As both (experimental and AI-based) approaches ultimately are validated experimentally, we prefer a clear distinction in our manuscript. We replaced traditional with classical for consistency. This is also consistent with the abstract: “While both classical and AI-based structural approaches have recently provided crucial insights”.

3. Line 122, it is not clear if the authors are talking about backbone interactions with the glycine residues that are mediate the antiparallel beta-sheet interactions? Have they considered passing their models through something like PISA to show the actual interactions involved in these beta-sheet interactions and thereby ligand binding? It would be nice to see some H-bonds and rough buried surface area calculations to support these observations.

Our response: We thank the reviewer for this suggestion. We added a new Suppl. Fig. 3, which depicts the putative H-bonds formed as well as the predicted distances for both the RLP23 and RLP42 tripartite complex (PISA analysis results). We revised the manuscript accordingly.

Considering the RLP23-nlp20-BAK1 prediction, the buried surface area of nlp20 is 1569.6 \AA^2 , which is 61.5% of total solvent-accessible area. In addition, Gly677 and Gly679 were completely buried in the interface, supporting that the side chains of Y4 and W6 interact on top of the glycine residues. There are seven hydrogen bonds between β -strands of RLP23 and nlp20, clearly showing β -sheet formation.

Considering the RLP42-pg13-BAK1 prediction, 67.7% of solvent accessible area of pg13 (1040.3 \AA^2) is buried in the interaction interface. Six hydrogen bonds between β -strands of RLP42 and pg13 are found.

- Related to this is the statement in line 143 indicating that their approach to mutate G677, G699 was to destabilise interactions with Y4 and W6 – would be good to give the reader an idea of how these residues stabilise the interaction in the first place as it is not obvious from figure 1 up until this point in the manuscript.

Our response: We thank the reviewer for this suggestion and adapted the text accordingly. As discussed above, the novel Suppl. Fig. 3 will now also help the reader to understand that the addition of two bulky hydrophobic acids would create steric hindrance and therefore destabilize the interaction between the ligand and the ID.

4. Figure 1, color scheme is poor. We would recommend making the ligand yellow throughout and simply highlighting the residues of interest. We realise this is not possible for the receptor due to the involvement of the glycine so color changes make sense here.

Our response: We thank the reviewer for this suggestion. We adapted Fig. 1 as suggested and adapted the legend accordingly. Similarly, we adapted this color change throughout all figures for consistency.

5. Section 175-189, these are not results. This section seeks to provide a structural context (based on the models presented here) to previously identified and studied mutations. Ultimately it is an interpretation based on a structural model, with no additional data added. It does not belong in the results. In many ways this could all be summarised in supplementary data as a table and perhaps some discussion dedicated to this.

Our response: We thank the reviewer for this suggestion. We moved the corresponding sentences from the result section to the legend of Suppl. Fig. 6.

6. Line 301, use of the word architectural appears to be out of context

Our response: We thank the reviewer for this suggestion; we agree and have changed the sentence accordingly.

7. Line 377, the statement is a little strong given this mutation in RLP42 can still recruit BAK1 (Fig. 4E)

Our response: We thank the reviewer for this suggestion; we agree that ‘impair’ is too vague a description (ranging from affected to abolished) and have therefore revised the sentence.

8. The island domains (IDs) of the LRR-RLPs and the subfamily Xb-LRR-RKs are different in terms of the presence of the lysine-containing motifs as per the authors, thereby having different roles in BAK1 recruitment. Has there been any work done to show how different orthologues of the BAK1 co-receptors are recruited by these two classes of LRRs? If not, would this be an angle worth exploring for the authors in support of one of their discussion points in line 316 on ‘comparative genomics and reevaluating existing variants on both sides of the interaction’

Our response: We thank the reviewer for this suggestion. The current understanding of the role of BAK1 and related SERK proteins as co-receptors for LRR-RKs and LRR-RPs suggests that SERKs are most likely biochemically equivalent with reported different genetic contributions potentially linked to different expression patterns; although some receptors might have some preference for some SERKs over others (e.g. FLS2 vs. EFR that seems to favor BAK1; Roux et al., *Plant Cell* 2011). Accordingly, all LRR-RPs characterized within our study, RLP23, RLP42 and RLP32, were shown to associate with multiple SERKs upon ligand treatment (Albert et al., *Nature Plants* 2015; Fan et al., *Nature Comm.* 2022; Zhang et al., *Nature Plants* 2021). As such, we are not aware of any study reporting differential recruitment of distinct SERKs by LRR-RKs or LRR-RPs.

Response to Reviewer 2

Our response: We thank the reviewer for their contribution to the review process and the positive feedback.

Response to Reviewer 3

This paper seeks to use an AI/computational approach (AlphaFold) to better understand how plant immune receptors bind to their ligands. Here the authors find that several members of the RLP subclass of LRR receptors (lacking an intracellular kinase domain) have a non-LRR “island domain” embedded within the LRR, and can “loop out” of the typical LRR horseshoe fold. This domain has been previously shown to be involved in substrate binding by x-ray crystallography. In three different RLP-ligand systems, AlphaFold satisfyingly predicted the ID was involved in ligand binding, and also coordinated the interaction of the RLP/ligand complex with BAK1, the kinase protein, required for downstream signaling. These hypothetical structures were then validated by mutagenizing residues (often conserved) that were located in key interaction surfaces. In some cases the mutations were pre-existing in the literature, and the structures allowed for the opposing residues to also be probed for function. The authors went on to test both a variety of downstream signaling outputs, as well as assessing interaction via co-immunoprecipitation. In general the loss of interaction via coIP correlated well with loss of signaling assuming that we can invoke different thresholds for a loss-of-function in the various assays (i.e. how much loss of interaction is expected to result in a loss of signaling? – hard to say).

In general, the paper is clear and well-written. I appreciated that the authors clearly stated that these models were hypotheses to be tested (and then did so). It would have been nice to see the authors really nail down the reality of what is going on by making compensatory receptor/ligand mutations that are loss-of-function alleles individually, but can function when combined together. But there is no guarantee that this approach is always possible, and I’m satisfied with the evidence presented. It is nice to see the authors include the two related ligands, and also a second ligand class, to show that the approach is at least somewhat generalizable.

This story should be of wide interest to the plant immunity field, and potentially beyond to other LRR-containing proteins across the tree of life.

There are a few minor points and graphical changes that could improve readability that I will list below. I don't have any major concerns, nice story.

We thank the reviewer for the positive feedback and appreciation of our work.

Individual points:

It would be extremely useful to renumber the amino acid sequences of the PDB files so that they match the figures. If not, every reader has to go dig up the full-length RLP sequences and do it themselves to try to reconcile them with the figures.

Our response: We thank the reviewer for this suggestion. We agree and have changed the PDBs accordingly. We also noticed that we originally uploaded an incorrect PDB for the prediction of RLP42-pg13; this is now corrected as well as the ipTM values shown in Fig. 1A.

Figures: I think that the main figures would benefit from some representation of where we are on the complex. This "zoomed-out" big picture is indeed in the supplemental, but it'd be really useful for the reader to understand this without digging. If no space, fine.

Our response: We thank the reviewer for this suggestion. Considering space and figure organization, we opted to keep the entire predictions in the supplementary. However, we did add spatial indications to the zoom-in figures, to make it easier for the reader to orientate themselves.

Figure 3A: I found this figure not that helpful on its own for understanding the modeled complex. The main text is also vague on which residues are actually interacting, how far they are apart, etc. I'd consider color-coding the important residues by protein. Would be nice to have a zoom-in on the proposed ionic interactions to show how they are oriented and that the distances are reasonable. I don't think that the space-filling overlay is useful here, and it adds clutter to an already complex image (also its not used in the other figures).

Our response: We thank the reviewer for this suggestion. Consistent with Fig. 1 (as requested by Reviewer 1), we now also used the same color-code in Fig. 3 for the important residues, *i.e.* yellow for the ligand and light grey for the receptor. The legend was adapted accordingly.

Additionally, we removed the space-filling overlay, except for the original zoom-in as it helps highlighting the binding pocket of F653. Finally, we added a zoom-in that focuses on the proposed ionic interactions and added distances between the predicted interactions, as suggested.

Fig 3 B-D; The presentation of statistics is a bit confusing (* for some, and p-value for others even though the test is the same?). If you could color-code these and/or add more explanatory text to the legend it would be much more reader friendly.

Our response: We thank the reviewer for this suggestion. We now have used a color-code (across all figures) and adapted the legends accordingly.

Figure 4: I think that this figure is not making clear what the tyrosine is doing. You could add a zoom in that shows the tyrosine relative to the opposing beta-sheets, or similar to clarify. As is, I get a better understanding of the charged residues, while the title of the figure focuses on the tyrosine. Similarly to 3A, you could color-code residues to make it more obvious what was going on (e.g. would be more clear that K665 is on the black ID loop, and E727 is on the LRR).

Our response: We thank the reviewer for this suggestion and added a zoom-in that focuses on the role of the tyrosine and the putative interaction with the opposing beta-sheets of the LRR-backbone. Besides, we used a color-code, as suggested for E727, as coloring of the ID residues in black is suboptimal due to the black beta-sheets in the background.

Figure 4A: K655 is typo, should be K665.

Our response: We thank the review for spotting this typo, we corrected the label in Fig. 4A.

ColPs: An empty vector control on the input blots to show that the bands are indeed what we expect them to be would have been nice.

Our response: We unfortunately do not have these controls but are confident that the indicated bands correspond to the correct proteins, based on previous studies and experiments.